# Roscovitine, a CDK Inhibitor, Reduced Neuronal Toxicity of mHTT by Targeting HTT Phosphorylation at S1181 and S1201 In Vitro

**DOI:** 10.3390/ijms252212315

**Published:** 2024-11-16

**Authors:** Hongshuai Liu, Ainsley McCollum, Asvini Krishnaprakash, Yuxiao Ouyang, Tianze Shi, Tamara Ratovitski, Mali Jiang, Wenzhen Duan, Christopher A. Ross, Jing Jin

**Affiliations:** 1Division of Neurobiology, Department of Psychiatry, Johns Hopkins University School of Medicine, Baltimore, MD 21287, USAyouyan12@jh.edu (Y.O.); tratovi1@jhmi.edu (T.R.); wduan2@jhmi.edu (W.D.); caross@jh.edu (C.A.R.); 2Department of Neurology, Neuroscience and Pharmacology, Johns Hopkins University School of Medicine, Baltimore, MD 21287, USA; 3Department of Pharmacology and Molecular Sciences, Johns Hopkins University School of Medicine, Baltimore, MD 21287, USA

**Keywords:** Huntington’s disease, roscovitine, CDK5, post-translational modification (PTM), toxicity

## Abstract

Huntington’s disease (HD) is an autosomal dominant neurodegenerative disease caused by a single mutation in the huntingtin gene (HTT). Normal HTT has a CAG trinucleotide repeat at its N-terminal within the range of 36. However, once the CAG repeats exceed 37, the mutant gene (mHTT) will encode mutant HTT protein (mHTT), which results in neurodegeneration in the brain, specifically in the striatum and other brain regions. Since the mutation was discovered, there have been many research efforts to understand the mechanism and develop therapeutic strategies to treat HD. HTT is a large protein with many post-translational modification sites (PTMs) and can be modified by phosphorylation, acetylation, methylation, sumoylation, etc. Some modifications reduced mHTT toxicity both in cell and animal models of HD. We aimed to find the known kinase inhibitors that can modulate the toxicity of mHTT. We performed an in vitro kinase assay using HTT peptides, which bear different PTM sites identified by us previously. A total of 368 kinases were screened. Among those kinases, cyclin-dependent kinases (CDKs) affected the serine phosphorylation on the peptides that contain S1181 and S1201 of HTT. We explored the effect of CDK1 and CDK5 on the phosphorylation of these PTMs of HTT and found that CDK5 modified these two serine sites, while CDK5 knockdown reduced the phosphorylation of S1181 and S1201. Modifying these two serine sites altered the neuronal toxicity induced by mHTT. Roscovitine, a CDK inhibitor, reduced the p-S1181 and p-S1201 and had a protective effect against mHTT toxicity. We further investigated the feasibility of the use of roscovitine in HD mice. We confirmed that roscovitine penetrated the mouse brain by IP injection and inhibited CDK5 activity in the brains of HD mice. It is promising to move this study to in vivo for pre-clinical HD treatment.

## 1. Introduction

Huntington’s disease (HD) is an inherited autosomal dominant neurodegenerative disorder. Genetic etiology was discovered decades ago [1]. However, there is still no cure. An average person has two alleles of the huntingtin gene (*HTT*), with CAG trinucleotide repeats at its N-terminal not exceeding 36. An individual bearing CAG repeats exceeding 40 will develop HD symptoms, including motor deficit, cognitive decline, and psychiatric perturbation. Most HD patients start to show HD symptoms around their 40s, while some juvenile HD patients with exceptionally long CAG repeats will have an early onset. Mutant huntingtin gene (*mHTT*) with CAG repeats longer than 40 encodes mutant HTT (mHTT) protein, which has an expanded polyglutamine tracts (polyQ) on its N-terminal [2,3,4]. Although RNA antisense transcription is believed to be involved in HD toxicity [5], mHTT toxicity appears to be polyQ length-dependent. Mutant HTT prominently affects the striatum of the brain, causing brain atrophy and resulting in a reduced volume of the striatum and cortical and other brain regions [2,3,6]. The mechanism of mHTT toxicity includes mitochondrial dysfunction, a deficit of axonal transportation, abnormal transcription and RNA splicing, etc., which are believed to be due to the abnormal function of mHTT or/and the loss of function of normal HTT [2,3,6,7]. After 30 years of gene discovery, HD still has no cure. Therapeutic approaches include small molecules to reverse the toxic consequences of mHTT, gene therapy to correct the *mHTT* gene in different cell types, and lowering the *mHTT* expression levels of mRNA, protein, or patients’ specific single nucleotide polymorphisms (SNPs). However, none of those strategies has succeeded so far.

The *HTT* gene contains 67 exons and encodes a large protein with a molecular weight of about 350 KD. HTT is a scaffold protein with roles in many cellular functions, including cellular transport, endocytosis, and gene transcription, as well as stress response [8,9,10,11,12,13,14,15,16,17,18,19,20,21,22,23]. The pathological mechanisms of HD are believed to involve predominantly a gain of toxic function of mHTT at RNA and/or protein levels, resulting in abnormal transcription, mitochondrial dysfunction, oxidative stress, abnormal metabolism, abnormal axonal transportation, protein dislocation, and impaired protein clearance [16,24,25,26]. There is no disease-modifying treatment for HD [10]. Therapeutic strategies have been developed to lower *mHTT* expression, reduce mHTT toxicity, increase mHTT clearance, or recover the pathways disrupted by mHTT. Therapeutic targeting of the root cause of HD to lower *mHTT* is promising [22,27,28,29,30,31,32] but has many potential problems, including the use of large molecules with poor brain penetration [33,34,35]. The unsuccessful clinical trials of recent *mHTT* lowering encouraged the use of small molecules to reduce mHTT toxicity as an alternative approach to developing HD therapies.

Our group and others have identified numerous post-translational modification sites (PTMs) on HTT protein using animal models and postmortem human brain samples [36,37,38,39,40,41]. These modifications include phosphorylation, acetylation, arginine methylation, SUMOlation, palmitoylation, etc. PTMs can affect HTT structure, protein-protein interactions, function, and mHTT toxicity. The N-terminal PTMs within exon 1 have been extensively studied, while not so much is known about the PTMs outside of exon 1. Among modifications, phosphorylation has been studied the most, including the S13, S16, S116, and S421 sites [36,37,39,40,41,42]. Enzymes/kinases catalyzing these modifications have been targeted to develop therapeutic interventions. It has been shown that modifying these PTMs can ameliorate mHTT toxicity [36,37,39,40,41,42], indicating that PTMs could be potential therapeutic targets for HD.

To search for potential kinases that can affect HTT PTMs, we performed an in vitro kinase assay using HTT peptides bearing the PTMs of HTT that we have previously identified by mass spectrometry [37], including S1181 and S1201. We screened a kinase library including 368 kinases (the list of kinases is included in Appendix A) and found that these two serine sites are predominantly targeted by cyclin-dependent kinases (CDKs), especially by CDK1 and CDK5.

CDKs are Ser/Thr protein kinases that associate with specific cyclin subunits in order to be activated. There are more than 13 family members [43], which are all phosphorylate Ser-Pro or Thr-Pro sites, with a preference for the basic residues Lys and Arg at proximal upstream and downstream positions. CDKs play indispensable roles in specific phases of the cell cycle. CDK5, a family member of the cyclin-dependent kinases, plays a pivotal role in the central nervous system.

CDK5 is specially expressed in the brain and is indispensable for brain development. In the adult brain, it is essential for numerous neuronal processes, including higher cognitive functions such as learning and memory formation [44,45]. However, Cdk5 activity becomes deregulated in several neurological disorders, such as Alzheimer’s disease, Parkinson’s disease, and Huntington’s disease, which leads to neurotoxicity. Therefore, precise control over Cdk5 activity is essential because of its physiological functions.

Unlike other CDKs which are activated by different cyclins, CDK5 is activated by the coactivators p35 and p39 in the brain. CDK5 is also a unique CDK specifically expressed in the brain. CDK5 is indispensable for embryonic brain development [46,47,48]. In adults, it has a critical role in various neuronal functions [44,45,49,50,51,52]. It is involved in regulating neuronal survival [49], synaptic plasticity [51], learning and memory formation [44,45], pain signaling [50], drug addiction [52], and long-term behavioral changes [45,50,52]. Precise control of the activity of CDK5 is critical for normal brain function. Both CDK1 and CDK5 are involved in neurodegenerative disorders [53,54,55,56,57,58,59,60,61,62,63,64,65,66,67]; CDK5 dysregulation is evidenced in the pathogenesis of HD [44,56,63,68].

In the present study, we studied the effect of CDK5 on mHTT serine phosphorylation. We also evaluated the effect of recovering a well-known CDK5 inhibitor on CDK5 activity and the inhibition of S1181 and S1201, as well as its potential to be used for preclinical HD treatment.

## 2. Results

### 2.1. In Vitro Kinase Assays Identified Upstream Kinases That Modified HTT Peptides Phosphorylation at Serine Sites of 1181, 1201, and 2653

Kinase assays were performed in vitro by Reaction Biology Corp for 368 recombinant human kinases (Appendix A). In this study, six peptides, including S116, S116/S120, S421, S1181, S1201, and S2653 of the human HTT protein, were targeted by a broad range of kinases (Figure 1 and Appendix A). Among these kinases, CDKs specifically target the serines 1181 (S1181), 1201 (S1201), and 2653 (S2653) of HTT peptides (Figure 1). S1181 was targeted by both CDK1 and CDK5, while S1201 and S2653 were targeted by CDK1 only with a 5% cut-off threshold. It was interesting that none of the N-terminal serine sites (S116, S120, and S421) was targeted by CDKs in our assay (Appendix A). We then compared the amino acid sequence of HTT around these three serine sites among different species. The amino acid sequence surrounding the serine sites of 1181 and 1201 (but not S2653) are highly conserved in vertebrates (Figure 2A). The two serine sites, S1181 and S1201, share the consensus target sequence for CDK1 and CDK5. We further illustrated the CDKs involved in the phosphorylation of these two serines with a 1% cut-off threshold (Figure 2B,C). For CDKs involved in the phosphorylation of these three serine peptides, both CDK1 and CDK5 had relatively higher activity in the phosphorylation for S1181 and S1201 peptides among all tested CDKs, but not for S2653.

### 2.2. The Knockdown of CDK5, but Not of CDK1, Reduced the Phosphorylation Level of HTT S1181 and S1201

To explore the effect of CDK1 and CDK5 on the phosphorylation of HTT S1181 and S1201, HEK293 cells were transfected with pooled siRNA targeting human CDK5 or CDK1 for 16 h and then transfected with a plasmid expressing full-length mutant huntingtin with 82 polyglutamine (FL-82Q) for another 24 h. Then, cells were collected and processed for western blot analysis. CDK5 expression was knocked down by ~31% (Figure 3A,B). The levels of p-S1181 and p-S1201 were significantly reduced by the CDK5 knockdown (Figure 3C–F). Though CDK1 expression was reduced by ~50% after transfection with CDK1 siRNA, the levels of p-S1181 and p-S1201 were not reduced (Appendix A), suggesting CDK1 may not play a role in the phosphorylation of S1181-HTT and S1201-HTT.

### 2.3. Overexpression of CDK5 Induced Phosphorylation of S-1181-HTT but Not S-1201-HTT

To further explore the effect of CDK5 on the phosphorylation of S1181 and S1201, a plasmid expressing human CDK5 was transfected into HEK293 cells together with the HTT plasmid FL-82Q. Twenty-four hours after transfection, cells were collected to test the level of phospho-S1181 (p-S1181) and S1201 (p-S1201) by western blot. The CDK5 expression level was determined after transfection (Figure 4A,B). The level of p-S1181 was significantly increased by CDK5 overexpression (Figure 4A,C,D), but not for S1201 Interestingly, the level of total huntingtin (2166) and mutant huntingtin (MW1) were increased by CDK5 overexpression (Figure 4A,E,F). Increased CDK5 expression was also seen in 4-month-old zQ175 HD mouse brain (Appendix A) in the primary cultured cortical neurons (Appendix A).

### 2.4. CDK5 Affects the Cell Toxicity of mHTT

The phosphorylation of S1181 was induced by CDK5, indicating that CDK5 may also affect the mHTT-induced neuronal toxicity. To explore the effect of CDK5 on mHTT toxicity, we utilized SThdh^Q111/Q111^ cells expressing human *mHTT* with 111 polyQ repeats. SThdh^Q111/Q111^ cells undergo cell death upon withdrawal of the nutrients from the culture medium. We transfected SThdh^Q111/Q111^ cells, as well as SThdh^Q7/Q7^ cells (expressing *HTT* with normal polyQ size) with pooled siRNA targeting mouse CDK5 or a plasmid expressing human CDK5. Twenty-four hours after transfection, the serum in the medium was withdrawn and cells were cultured for another 24 h. Cell death was analyzed using CytoTox-Glo Cytotoxicity assay. The overexpression of CDK5 increased cell death, while the knockdown of CDK5 significantly reduced cell death in SThdh ^Q111/Q111^ cells compared with the control group (Figure 5).

### 2.5. Modification of S1181 and S1201 Sites Altered the Toxicity of mHTT

Modulation of CDK5 levels affected the phosphorylation of S1181 and S1201 and mHTT-induced toxicity. To investigate the role of the serine site of 1181 and 1201 in mHTT-induced cell toxicity, we utilized the FL-82Q HTT plasmid with a serine site altered to phospho-null amino acids (S → A). This allowed us to verify the effect of modifying S1181 and S1201 on mHTT toxicity. HEK293 cells were transfected with constructs of full-length mHTT (FL-82Q) and FL-82Q with serine sites modified to alanine (A) as indicated in the figure. Cell death was evaluated by caspase3/7 activity using a Caspase-Glo 3/7 assay. The modification of S1181A and S1201A induced significantly less cell death vs. unmodified HTT FL-82Q (Figure 6A). The effect of modifying serine at the site of 1181 and 1201 on mHTT toxicity was further confirmed in the primary neuronal HD cell model as we published in earlier works [36,69,70], where primary cultured cortical/striatal neurons were co-transfected with GFP and constructs expressing full-length huntingtin with either 23Q (FL-23Q) or 82Q (FL-82Q) or modified 82Q (FL-S1181A, FL-S1201A) (Figure 6B). Both S1181 and S1201 modification affected mHTT toxicity (Figure 6B).

### 2.6. Roscovitine, a CDK Inhibitor, Protected Cells from mHTT-Induced Cell Toxicity

Roscovitine is a well-known CDK inhibitor. It inhibits the activity of CDKs, including CDK5, CDK1, and other CDKs [66,71,72,73,74]. To test the effect of roscovitine on mHTT-induced toxicity, immortalized striatal cells (SThdh ^Q111/Q111^) were stressed by serum withdrawal and treated with roscovitine for 48 h. The dead and live cells were measured using a CytoTox Glo cytotoxicity assay. Cell toxicity was evaluated by the ratio of dead/live cells. SThdh ^Q111/Q111^ cells underwent cell death when the serum was withdrawn from its culture medium. Roscovitine treatment significantly reduced cell death in a dose-dependent manner compared to the vehicle-treated group (Figure 7A). We also tested the effect of roscovitine on mHTT toxicity in the primary neuronal HD cell model, as published in our previous works [36,69,70]. Primary cultured cortical/striatal neurons were co-transfected with GFP and constructs expressing full-length huntingtin with either 23Q or 82Q, and then treated with roscovitine. After 48 h of treatment, neurons were fixed, and nuclei were stained with Hoechst. Cell death was evaluated by nuclei condensation assay and presented as a fold of toxicity induced relative to the full-length-23Q (FL-23Q) group. Roscovitine protected neurons from mHTT-induced neuronal toxicity (Figure 7B).

### 2.7. Roscovitine Mitigated the Phosphorylation of S1181 and S1201 of mHTT by Inhibiting CDK5 Activity but Not CDK1

To explore whether roscovitine affects the phosphorylation of S1181 and S1201 of mHTT, HEK293 cells were transfected with mutant huntingtin expressing plasmid (FL-82Q) and treated with or without roscovitine for 48 h. Then, cell lysate was collected, and the expression of p-S1181-HTT and p-S1201-HTT were detected by western blot. Roscovitine significantly reduced the level of p-S1181 and p-S1201 of mHTT-expressing cells (Figure 8A,B,E,F). Roscovitine significantly reduced the level of CDK5 but not CDK1 (Figure 8A,B,G,H), suggesting that roscovitine reduced p-S1181 and p-S1201 by inhibiting CDK5.

### 2.8. Roscovitine Crossed the BBB and Inhibited CDK5 Activity in Mouse Brain

In order to verify if roscovitine is suitable for in vivo study, roscovitine was injected into 4-month-old WT mice with ascending doses of 25, 50, and 100 mg/kg by intraperitoneal (IP) injection. Each dose group had four or five mice. Plasma and brain samples were collected at 30 min, 1 h, 2 h, 4 h, 8 h, and 24 h after IP injection. Roscovitine concentration was measured using LCMS/MS. Roscovitine concentration reached a peak rapidly 30 min after injection and dropped rapidly as well (Figure 9A). High dose roscovitine (100 mg/kg) was intolerable for the mice. Half of the mice injected with a high dose were dead within 20 min of the injection. So, we used two lower doses of roscovitine, 25 mg/kg and 50 mg/kg, to further explore the tolerability of roscovitine by subchronic injection in HD mice. Roscovitine was injected into 4-month-old zQ175 HD mice with 25 mg/kg and 50 mg/kg by IP daily for 3 weeks, using littermate mice as control. Twenty-four hours after the last injection, brain samples were collected. CDK5 activity in the HD mouse brain was measured using a commercial kit. Three weeks of roscovitine administration reduced CDK5 activity in the HD mouse brain compared with the vehicle-injected HD mouse brain (Figure 9B), while it did not change the CDK5 activity in the brain of wild-type mice (Appendix A).

### 2.9. Roscovitine Was Safe and Tolerable in HD Mice

Three weeks of IP injection did not affect the body weight of the zQ175 HD mice (Figure 10A) or the behaviors measured by CatWalk (Figure 10B). To evaluate whether the roscovitine injection caused a glial reaction, cortical samples from the injected HD mice were prepared, and western blot analysis was performed to detect the expression of neuronal and glial markers in the brain. The pan-neuronal marker (NeuN) and two glial cell markers, GFAP (astrocyte) and IBA1(microglia), were used to indicate the cellular responses in the mouse brain to the roscovitine injection. There were no significant differences between the vehicle- and roscovitine-treated mice in terms of the expression of NeuN, GPAF, IBA1, and mHTT (detected with MW1 antibody) (Figure 11).

## 3. Discussion

In this study, we found that S-1181 of mHTT is a target of CDK5. Roscovitine inhibited CDK5 activity and reduced mHTT-induced toxicity in different in vitro HD cell models. Roscovitine is a well-known CDK inhibitor. It reversed the phosphorylation of S1181 of mHTT through inhibiting CDK5. We further confirmed that roscovitine penetrated the mouse blood–brain barrier (BBB) by intraperitoneal (IP) injection and inhibited the activity of CDK5 in HD mice brain. It is worth testing the effect of roscovitine for the longer term in HD mice to reduce mHTT toxicity and improve the outcome of HD neuropathology.

Huntingtin is a large protein without enzymatic activity or small molecule binding sites within HTT protein, which makes it difficult to target using small molecules. However, HTT has multiple post-translational modification (PTM) sites that can modulate mHTT toxicity [23,36,37,40,41]. Our group, together with other groups, identified numerous PTM sites in HTT and confirmed their roles in modulating mHTT toxicity [36,37,39,40,41,42]. These PTMs can be modified by phosphorylation, acetylation, methylation, sumoylation, palmitoylation, etc. [23,36,37].

Among these PTM sites, the phosphorylation of serine at sites 13, 16, 116, 120, and 421 can affect their structure configuration and modulate the toxicity of mHTT [36,37,39,42]. However, there is controversy about the effect of phosphorylation of the serine 1181 and 1201 sites on mHTT toxicity. Our present study indicates that inhibiting the phosphorylation of S1181 and S1201 protects cells from mHTT-induced toxicity. While studies from Humbert group suggested that inhibiting the phosphorylation at S1181 and S1201 was toxic [65], our present study, together with others [75,76] suggest that inhibiting CDK5 activity is neuroprotective.

In our present study, we explored whether small molecules, such as kinase inhibitors, can reduce mHTT toxicity by affecting kinase activity that modifies the PTMs of mHTT [2,36,37,39,41]. To explore whether small molecules can be used to alter mHTT toxicity by targeting the PTMs in mHTT, we screened a kinase library containing 368 kinases using synthesized huntingtin peptides bearing different serine sites of HTT, including S116, S120, S421, S1181, S1201, and S2653. Each site can be targeted by multiple kinases, and a single kinase can target multiple serine sites as well (Figure 1, Figure 2 and Appendix A). Among these kinases, CDK1 and/or CDK5 targeted the serines outside exon 1 of HTT, including S1181, S1201, and S2653 (Figure 1). We focused on CDK5 and CDK1 because these two CDKs, unlike other CDKs, are involved in neurological disorders [54,55,57,77]. Notably, S2653 does not have consensus sequences for CDK targets, while S1181 and S1201 bear a consensus sequence for both CDK5 and CDK1 phosphorylation. We hypothesized that knocking down CDK5 and CDK1 would reduce the levels of S1181 and S1201. Indeed, in the CDK5 knockdown cells, but not in the CKD1 knockdown cells, the levels of p-S1181 and p-S1201 were reduced (Figure 3 and Appendix A), indicating that CDK5 mediates the phosphorylation of S1181 and S1201.

The overexpression of CDK5 in cells expressing FL-82Q increased the level of p-S1181 but not S1201. This is consistent with the in vitro kinase assay in which CDK5/p25 dramatically phosphorylated S1181-HTT but not S1201. However, we did not expect that mHTT and total huntingtin levels would be increased by the overexpression of CDK5. This indicated that CKD5 over-activation may compromise the mHTT degradation in a pathway that requires further investigation.

Adequate CDK5 activity in the brain is critical for normal brain function. Over-activated CDK5 contributes to HD pathogenesis and other neurodegenerative disorders [56,62,63,75,78,79,80]. While other CDKs are not expressed or active in the brain, CDK5 is ubiquitously expressed in the brain and plays a critical role function in brain development and in the postmitotic neurons of an adult’s brain [43]. We found that the expression of CDK5 was altered in HD primary neurons and zQ175 HD mouse brain (Appendix A). Research has indicated that inhibiting CDK5 in HD mice had beneficial effects [75,76]. In the present study, knocking down CDK5 had a protective effect, and the overexpression of CDK5 induced severe cell death in mHTT-expressing cells (Figure 5), suggesting that reducing CDK5 activity could provide protection against HD. CDK5 is unique in the maintenance of normal brain function in the adult brain. CDK5 activation needs the coactivator of p35 and/or P39 in the brain. p35 and p39 can be cleaved into p10 and p25/p29, and p10 has a neuroprotective effect, while p25/p29 may activate CDK5, resulting in hyperactivity [43]. Activated CDK5 also can autophosphorylate p35 and induce its degradation. So, precisely controlling the activity of CDK5 in the adult brain is critical for normal function.

Our data indicated that S1181 and S1201 can be phosphorylated by CDK5 and inhibiting the kinase activity of CDK5 in vitro by siRNA or by roscovitine can reduce the toxicity of mHTT. mHTT with S1181A and S1201A alterations was less toxic compared with unaltered mHTT, supporting the idea that modifying the phosphorylation of S1181 and S1201 of mHTT could provide protection against HD. Small-molecule rescovitine, a CDK inhibitor, inhibited the phosphorylation of S1181 and S1201 of mHTT and reduced mHTT-induced cell death in HD cell models, indicating that the protective effect of roscovitine could be at least partially by inhibiting the phosphorylation of S1181 and S1201.

Roscovitine has been used in clinical trials for other diseases, especially in cancer research. In cancer-related clinical trials, roscovitine had such obvious side-effects that the trials were terminated. However, in the context of HD pathogenesis, since CDK5 may have critical roles for normal brain function, it would be better to maintain a certain level of CDK5 activity rather than totally inhibiting its activity, like in cancer treatment.

We assume that partially inhibiting the CDK5 activity in the context of HD will provide benefits by reducing the phosphorylation of S1181 of mHTT and thus reducing toxicity. To explore whether roscovitine administration can reach the mouse brain and inhibit CDK5 activity, we injected roscovitine into young mice by IP. Studies have shown that roscovitine can get into the mouse brain by IP and oral administration [81]. Our acute injection experiment indicated that roscotine can cross the blood–brain barrier of mice. However, the concentration of roscovitine reduced rapidly, which is consistent with other findings [82,83,84]. We also noticed that a high dose of roscovitine (100 mg/kg) was toxic to mice. We lost more than half of the mice from this group. The activity of CDK5 in the brain of HD mice was reduced by 25–30% by the subchronic administration of lower doses for three weeks by IP, which there was no difference between the two doses used in the present study. The subchronic administration of roscvitine did not alter the body weight of the HD mice, nor did it reactivate glial responses in the brain. Therefore, we suggest using a dose of 25 mg/kg for long-term injection in HD mice.

A lower dose of roscovitine could be beneficial to HD mice due to it partially preserving the activity of CDK5 in the brain. The present study is proof of the concept that small molecules like roscovitine can be used to target PTMs to reduce mHTT-induced toxicity in vivo.

## 4. Materials and Methods

### 4.1. In Vitro Kinase Assay

The in vitro kinase assay was carried out by the Reaction Biology Corp in the radiometric HotSpot™ assay format based on the transfer of ^33^P-labelled phosphate from ATP to the kinase substrate. Proteins and peptides in the HotSpot™ assay are captured via spotting of the reaction mix on a filter membrane. A kinase library containing 368 recombinant human kinases was screened using synthetic HTT peptides containing indicated PTM sites as substrates. Peptides were incubated with 368 kinases at a single dose of 20 µM with duplicates. The control compound, Staurosporine, was tested in 10-dose IC50 mode with 4-fold serial dilution starting at 20 or 100 μM. Reactions were carried out at 10 μM ATP. Data were presented as % enzyme activity (relative to RBC substrate, which is considered as 100%). Curve fits were performed where the enzyme activities at the highest concentration of compounds were less than 65%.

A total of six peptides were included, which were S116, S116/S120, S421, S1181, S1201, and S2653 of the human HTT protein. Huntingtin peptides harboring amino acid serine at sites 116 (S116), 421 (S421), 1181 (S1181), 1201 (S1201), and 2653 (S2653) were synthesized by GenScript.

### 4.2. Cell Culture

All the cell culture supplies were obtained from Corning, and the cell culture medium and supplements included were purchased from ThermoFisher Scientific ThermoFisher Scientific, Waltham, MA, USA) unless otherwise specified.

*Human embryonic kidney 293 cells (HEK293) cells* were obtained from Invitrogen (Waltham, MA, USA) and were grown in DMEM (4.5 g/L glucose) supplemented with 10% FBS, 100 mg/mL Geneticin, 100 units/mL penicillin, and 100 units/mL streptomycin at 37 °C/5% CO_2_.

*Immortalized striatal precursor cells* expressing normal *HTT* (STHdh^Q7/Q7^) or mutant *HTT* (STHdh^Q111/Q111^) were kindly provided by Dr. Marcy MacDonald and were prepared as described previously [85]. The cells were maintained at 33 °C in DMEM (4.5 g/L glucose) containing 10% FBS, and 400 g/mL G418, in a humidified atmosphere of 95% air and 5% CO_2_.

*Primary cortical and striatal mixed neurons* were prepared using the method we previously published [36,86]. All the experimental protocols involving animals were approved by the Johns Hopkins Institutional Animal Care and Use Committee (Protocol # MO21M315) and followed the National Institutes of Health guidelines for the use of experimental animals. Pregnant CD1 mice were purchased from Charles River Laboratories. Briefly, the day 17 pregnant mouse was anesthetized by leaving the mouse in a wide-mouth glass jar prefilled with isoplurane for 2 min. We used a 500 mL wide open glass jar with a tight-sealed lid as an induction chamber. The glass jar was separated into two parts by a fine mesh wire. Cotton soaked with isoflurane was put in the lower part of the glass jar, and the mouse stayed on the upper part of the jar to avoid direct contact with the isoflurane. A glass jar will help with observations of the mouse. All the procedures involved in the usage of isoflurane were conducted in a fume hood. Once the pregnant mouse had no response to tail pinching, the embryos were removed from the pregnant mouse by making a longitudinal incision in the low belly. The embryos were removed from the uterus and the embryos were decapitated and transferred to prechilled Hanks’ balanced salt solution (HBSS) buffer for further dissection. The pregnant mouse was euthanized by cervical dislocation after all the embryos had been removed. The neocortex and striatum of the embryos were dissected by removing the midbrain and hippocampus under a stereomicroscope. The cortical and striatal tissue was digested using trypsin for 15 min and then digested with DNase for 1 min. The tissue was dissociated by pipetting, and single-cell suspension was achieved by filtering the digested tissue through a cell strainer. Cells/neurons were plated at 5 × 10^4^ cells/well in 48-well plates pre-coated with poly-D-Lysine and laminin. The cells/neurons were maintained at 37 °C/5% CO_2_ in neurobasal medium containing 2% B27, 2 mM Glutamax, and 1% Pen/strep.

### 4.3. Plasmids and siRNA

Mammalian expression plasmids encoding full-length HTT of normal Q (FL-23Q) or expanded Q (FL-82Q) or expended Q with serine modification (FL-S1181A or FL-S1201A) were generated and sequenced by our lab as described previously [37]. Human CDK5 expressing plasmid was purchased from Addgene (Watertown, MA, USA; Addgene plasmid # 1346; a gift from Li-Huei Tsai). Pooled siRNA targeting human and mouse CDK5 and CDK1 was purchased from Santa Cruz Biotechnology (Santa Cruz Biotechnology, Inc., Dallas, TX, USA).

### 4.4. Transfection of Plasmid or siRNA

*Transfection of primary neurons.* Neurons were co-transfected using Lipofectamine 2000 (ThermoFisher Scientific) on day 6 in vitro (DIV) as per our previously published protocol [36,86]. Briefly, neurons were co-transfected with GFP and desired plasmids, namely, plasmids expressing full-length human huntingtin with either 23 (FL-23Q) or 82 (FL-82Q) poly-glutamine in exon 1, or FL-82Q-S1181A and FL-82Q-S1201A. Four hours after transfection, the neurons were switched to neurobasal medium addition with or without the tested compound. Forty-eight hours after transfection, the neurons were fixed with 4% paraformaldehyde in phosphate-buffered saline (PBS) for 30 min. Then, the nuclei were stained for 10 min using a solution of 0.2 μg/mL Hoechst 3342 (Millipore Sigma, Burlington, MA, USA) in PBS. The plate can be stored at 4 °C until further analysis. Cell deaths were analyzed using nuclei condensation assay, as published in our earlier works [36,70].

*Transfection of HEK293 cells.* HEK293 cells grew to 60–70 confluent, and then the culture medium was changed to OPTI-MEM serum-reduced medium. The plasmid or siRNA were transfected with Lipofectamine 2000 (ThermoFisher Scientific, Waltham, MA, USA) as recommended for HEK293 cells with DNA, i.e., with a Lipofectamine ratio of 1:2. After 6 h of transfection, the medium was changed back to growth medium and kept culture until further analysis, either for western blot or for cell toxicity analysis. Cell toxicity analysis was measured using the Caspase-Glo 3/7 assay kit (Promega, Madison, WI, USA) as recommended.

*Transfection of Immortalized striatal precursor cells.* STHdh^Q7/Q7^ and STHdh^Q111/Q111^ cells were transfected with siRNA targeting mouse CDK5 or a plasmid expressing human CDK5 using Lipofectamine 2000 as recommended. The cells were transfected for 24 h and we then withdrew the serum for another 24 h. Cell toxicity was measured by the CytoTox-Glo Cytotoxicity assay kit (Promega, Madison, WI, USA) as recommended.

### 4.5. Western Blot and Antibodies

The cells were homogenized in RIPA buffer (Millipore Sigma, Burlington, MA, USA) containing Halt (ThermoFisher Scientific, Waltham, MA, USA) and p-Halt (ThermoFisher Scientific, Waltham, MA, USA) plus PMSF (Cell Signal Technology, Danvers, MA, USA). Protein assay BCA kits (Thermo Scientific, Waltham, MA, USA) were used to determine the total protein concentration. A total of 25–50 ug protein was separated in a 4–20% gradient Criterion Tris-HCL gel (Bio-Rad, Hercules, CA, USA) and transferred to a PVDF membrane (Bio-Rad, Hercules, CA, USA). The membrane was blotted with the following primary antibodies: Anti-p-S1181 and Anti-p-S1201 generated by Seong Lab [37], MW1(Millipore Sigma), MAB2166 (Millipore Sigma), anti-CDK1 (Cell Signal Technology, Danvers, MA, USA), anti-CDK5 (Cell Signal Technology, Danvers, MA, USA), anti-NeuN (Millipore Sigma, Burlington, MA, USA), anti-GFAP (Invitrogen, Waltham, MA, USA), anti-IBA1 (FujiFilm, Valhalla, NY, USA), and anti-β-actin (Millipore Sigma, Burlington, MA, USA). After incubation with Horseradish peroxidase (HRP)-conjugated secondary antibodies, ECL luminescence substrates (Millipore Sigma, Burlington, MA, USA) were used to visualize the specific bands for each protein. The densitometry analysis was performed using Image J (ImageJ 1.53t).

### 4.6. Nuclear Condensation Assay

Cell toxicity experiments on the primary neurons were conducted as per the published protocol [36,70,86]. Automated picture acquisition was performed using a Zeiss Axiovert 200 (Zeiss, Hebron, KY. UAS) inverted microscope with a 20× objective, and mosaic images were obtained. Semi-automatic quantification of the nuclear intensity of the transfected cells was performed as before [36,70]. Cells were considered dead when their nuclear intensity was higher than the average intensity by two standard deviations. Each condition was performed in triplicate within each experiment, and each experiment was repeated in at least three independent neuronal preparations unless specifically indicated. Cell death was evaluated as toxicity induced compared with the normal Q group.

### 4.7. Roscovitine Preparation and Concentration Analysis

Roscovitine was purchased from Selleck Chemicals LLC. (Houston, TX, USA). For the cell culture, roscovitine was dissolved in dimethylsulfoxide (DMSO) to the concentration of 10 mM stocking solution and stored at −20 °C until usage. For the in vivo IP injection, first, roscovitine was prepared as 50 mg/mL stock solution in DMSO and stored at −20 °C. For the in vivo experiments, an injection solution of roscovitine was prepared freshly from the stock solution daily. The preparation of roscovitine should follow the order described here: 10% stock roscovitine, add 45% of 0.9% sodium chloride, mix well until clear, then add 45% PGE-400. It should be prepared immediately before performing the IP injection into a mouse. The roscovitine concentration was measured by Alliance Pharma Inc. (Malvern, PA, USA) by liquid chromatography–mass spectrometry/mass spectrometry (LCMS/MS).

### 4.8. CatWalk Analysis

We used CatWalk gait analysis system (Noldus, Netherlands system, Leesburg, VA, USA) to evaluate whether the roscovitine injection caused gait abnormalities, due to other measurements being unable to detect abnormalities in early HD mice [87,88]. The CatWalk gait analysis system consists of a black corridor on a glass plate with a green LED light inside. It is placed in a dark and silent room. Using the Illuminated footprints technology, the movement of paws is captured by a high-speed video camera positioned underneath the glass.

Before the test, mice were left in a light-cycle-reversed room for 3 weeks. On the day of the CatWalk test, the mice were placed in the testing room one hour prior to the test. All the tests were performed under dark conditions.

The walkway area was set up according to the company’s recommendations. The mice were then moved to the CatWalk XT and allowed to cross the corridor voluntarily for three accomplished runs. We placed a home cage at the end of the walkway as bait. A mouse walking across the runway without stopping, turning around, or changing direction will be considered as a compliance run. If a mouse turns around, it is set aside for 30 min and retested. Each mouse completes three compliant runs. The stride length was analyzed and represented as mean ± SEM.

### 4.9. Mice

Wild-type (Strain #: 000664) and heterozygous zQ175 mice (Strain #: 027410) were purchased from the Jackson Lab (Bar Harbor, ME, USA). Genotyping and CAG repeat size were determined by PCR of tail snips at Laragen Inc (Culver City, CA, USA). The CAG repeat length was 220 ± 3 in this batch of zQ175 mice. All the mice were housed under specific pathogen-free conditions and provided with food and water ad libitum. The study was approved by the Animal Care and Use Committee at Johns Hopkins University (protocol number: MO21M315) and carried out in strict accordance with the guidance for the Care and Use of Laboratory Animals of the National Institutes of Health.

### 4.10. Sample Collection

Plasma and brain samples from the injected mice were collected at different time-points and stored at −80 °C until further analysis. The plasma was collected using a K3-EDTA anticoagulant tube by gently mixing tubes and centrifuged for 10 min at 1000× *g* at 4 °C. Clear supernatant was collected into a new tube and stored at −80 °C for further analysis.

For the mouse brain sample collection, the mice were anesthetized by inhaling isoflurane for 2 min as described above. Once the mice had no response to tail pinching, they were perfused transcardially with 30 mL PBS to flush out the blood in the brain. Then, half of the brain was dissected for measuring the roscovitine concentration, and the other half was dissected for western blot analysis. The collected mouse brains were stored at −80 °C until usage.

### 4.11. CDK5 Activity Analysis

An ADP-Glo™ Kinase assay kit (Promega) was used to measure the activity of CDK5 as recommended by the manufacturer. Briefly, the brain tissues were homogenized using RIPA buffer without EDTA. The protein concentration was measured using a BCA kit (Thermo Scientific). A total of 50 ug protein was added for each sample. First, we diluted the enzyme, substrate, ATP, and inhibitor/samples in kinase buffer. Then, we added 1 µL of the inhibitor/sample, 2 µL of enzyme, and 2 µL of substrate/ATP mix into the wells of 384 low-volume plates and incubated them at room temperature for 10 min. Then we added 5 µL of ADP-Glo™ Reagent and incubated the samples at room temperature for 40 min. This was followed by adding 10 µL of kinase detection reagent and further incubating the samples at room temperature for 30 min. Then, we recorded the luminescence signal using a plate reader (Glo-Max Discover, Promega, Madison, WI, USA).

### 4.12. Statistical Analysis

Data were represented as mean ± SEM from triplicates. Statistical analysis was conducted using GraphPad Prism software version 8 (GraphPad, San Diego, CA, USA). One-way or two-way ANOVA or a two-tailed Student *t*-test was used to analyze the data according to the data type. The results were considered significant if the *p*-value was <0.05. Error bars indicate SEM in all the figures.

## Figures and Tables

**Figure 1 ijms-25-12315-f001:**
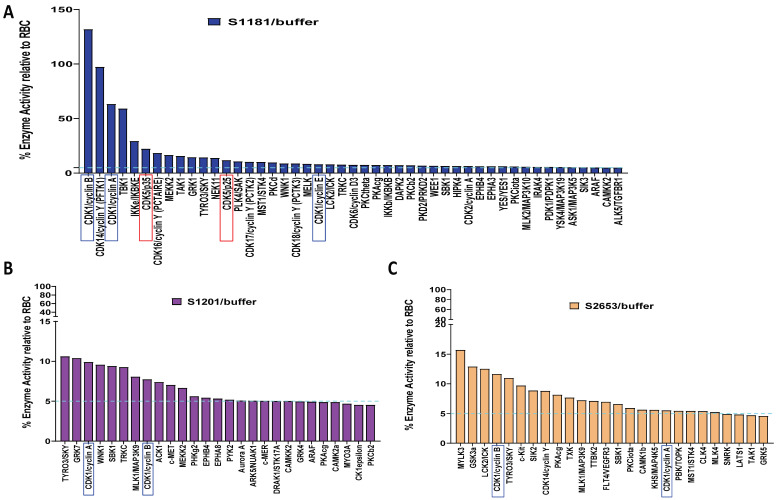
Kinase assays identified upstream kinases modifying the serine of HTT peptides in vitro. In vitro kinase assay was used to screen the kinase targeting HTT peptide bearing serine 1181, serine 1201, and serine 2653 of HTT. (**A**) Kinases targeting the serine 1181 of the HTT peptide. (**B**) Kinases targeting the serine 1201 of the HTT peptide. (**C**) Kinases targeting the serine 2653 of the HTT peptide. CDK1 is highlighted by blue rectangles and CDK5/p25 by red rectangles. RBC, red blood cell substrate. Dotted lines indicate the 5% cut-off of enzyme activity used for the kinase screening.

**Figure 2 ijms-25-12315-f002:**
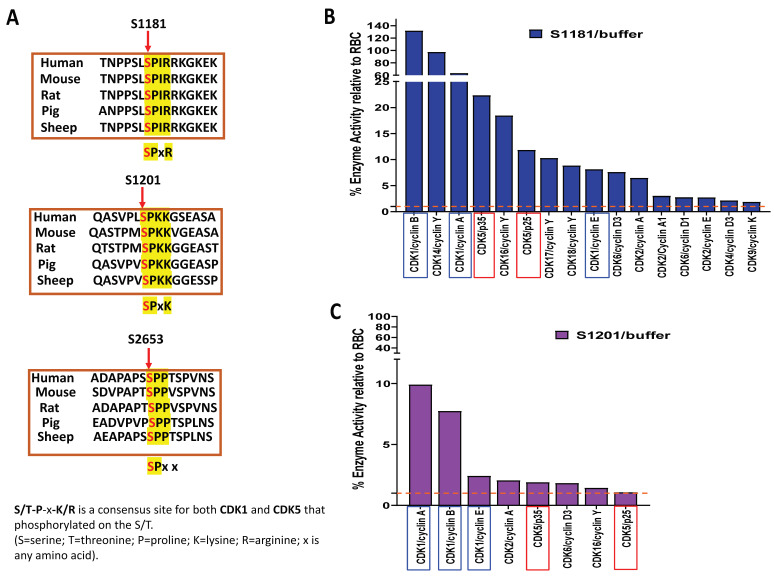
HTT peptides as potential as targets for CDKs. (**A**) The amino acid sequences surrounding serine of 1181, 1201, and 2653 of HTT from different species. S/T-P-x-K/R is a consensus site for both CDK1 and CDK5, which means that the amino acid of S or T followed by P-x-K/R could be phosphorylated by CDK1 and CDK5. (S = serine; T = threonine; P = proline; K = lysine; R = arginine; x is any amino acid). It is well conserved in vertebrates for S1181 and S1201 of the HTT sequence, but not S2653. (**B**) CDKs target serine 1181 of the HTT peptide. (**C**) CDKs target serine 1201 of HTT peptide. CDK1 is highlighted by blue rectangles and CDK5/p25 by red rectangles. RBC, red blood cell substrate. Dotted lines indicate the 1% cut-off of enzyme activity used for the kinase screening.

**Figure 3 ijms-25-12315-f003:**
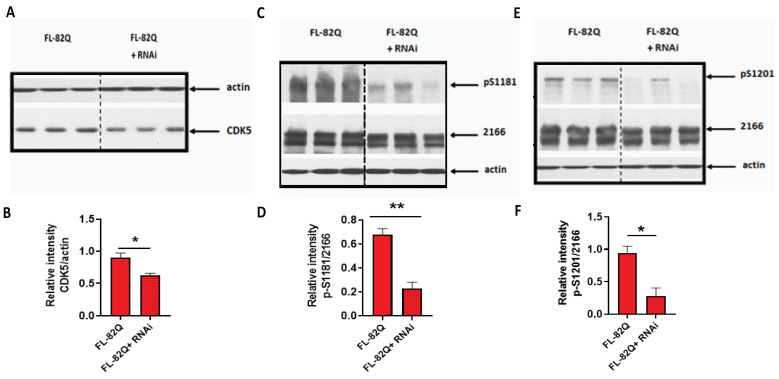
The effect of knocking down CDK5 on the phosphorylation of S1181 and S1201 of mHTT. HEK293 cells were co-transfected with a plasmid expressing full-length mutant huntingtin with 82Q (FL-82Q) and siRNA targeting human CDK5 for 24 h. The expression level of CDK5, phosphorylated S1181-HTT (p-S1181), and phosphorylated S1201-HTT (p-S1201) were detected by western blot. (**A**) The representative western blot for CDK5. (**B**) The quantification of CDK5 expression. (**C**) The representative western blot for p-S1181-HTT. (**D**) The quantification of p-S1181-HTT expression. (**E**) The representative western blot for p-S1201-HTT. (**F**) The quantification of p-S1201-HTT expression. The experiment was repeated by at least two different analysts. Each experiment had *n* = 3. One representative experiment is presented. 2166 is the anti-HTT antibody (MAB2166, Millipore). RNAi = pooled siRNA targeting human CDK5. A two-tailed Student *t*-test was used. * *p* < 0.05; ** *p* < 0.01.

**Figure 4 ijms-25-12315-f004:**
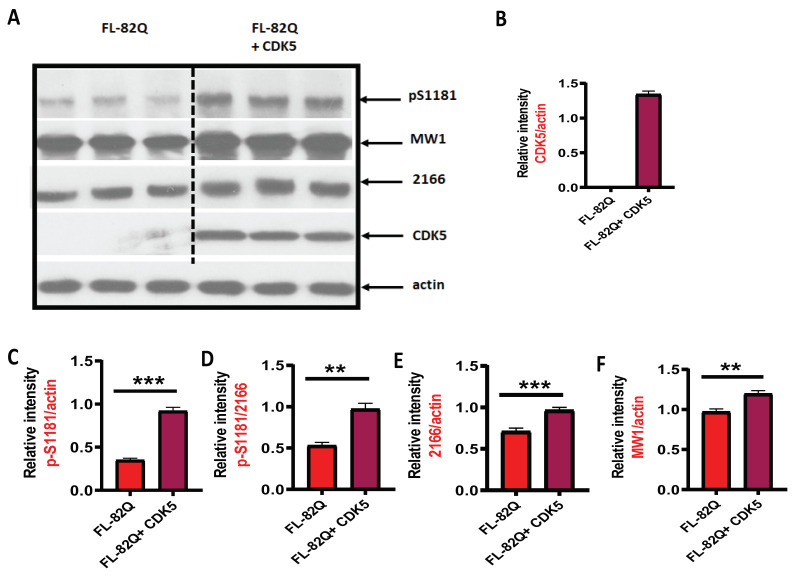
The effect of CDK5 overexpression on S1181 of mHTT. HEK293 cells were co-transfected with HTT plasmid FL-82Q and a plasmid expressing human CDK5 for 24 h. The expression levels of CDK5, p-S1181, and p-S1201 were detected by western blot. (**A**) The representative western blot for CDK5, p-S1181, 2166, and MW1. (**B**) The quantification of CDK5 expression. (**C**,**D**) The quantification of p-S1181 expression. (**E**) The quantification of total huntingtin (MAB2166) expression. (**F**) The quantification of mutant huntingtin (MW1) expression. The experiment was repeated by at least two different analysts. Each experiment had n = 3. One representative experiment is presented. 2166 is the anti-HTT antibody (MAB2166). MW1 is the anti-HTT antibody that binds to mutant huntingtin (clone MW1). RNAi = siRNA targeting human CDK5. A two-tailed Student *t*-test was used. ** *p* < 0.01; *** *p* < 0.001.

**Figure 5 ijms-25-12315-f005:**
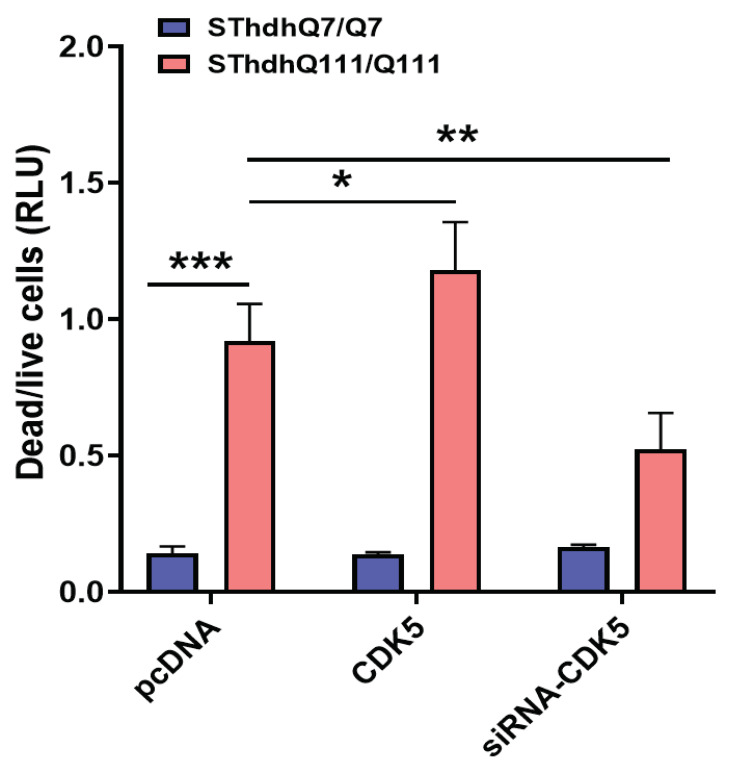
CDK5 level altered cell toxicity induced by mHTT. The effect of CDK5 overexpression or knocking down was evaluated in mouse striatal cells with different poly Q (SThdh ^Q7/Q7^ or SThdh ^Q111/Q111^). Cells were transfected with either a plasmid expressing human CDK5 or pooled siRNA targeting mouse CDK5 for 24 h. Cell death was measured with CytoTox-Glo cytotoxicity assay kit (Promega). The experiment was repeated by at least two different analysts. Each experiment had n = 3. One representative experiment is presented. Two-way ANOVA with Tukey’s multiple comparation was used for analysis. * *p* < 0.05; ** *p* < 0.01; *** *p* < 0.001.

**Figure 6 ijms-25-12315-f006:**
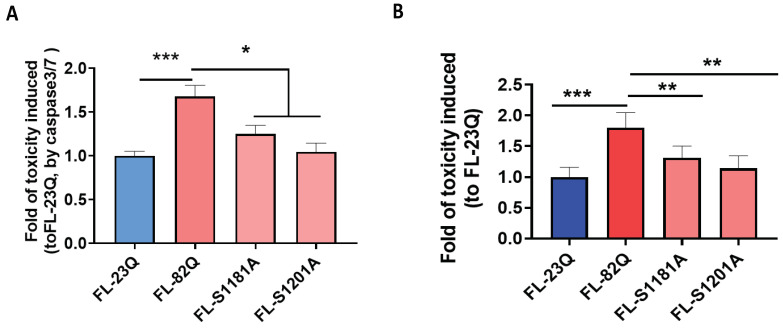
Phospho-null modification of S1181 and S1201 of mHTT altered the mHTT-induced cell toxicity. S1181 and S1201 in the plasmid of FL-82Q were artificially modified to alanine (A). (**A**) Modified plasmids were transiently transfected into HEK293 cells for 72 h, and cell toxicity was measured by caspase3/7 activity. (**B**) The modified plasmids were transient transfected in primary cortical neurons, and cell death was analyzed by nuclei condensation assay. The experiment was repeated by at least two different analysts. Each experiment had n = 3. One representative experiment is presented. One-way ANOVA with Dunnett’s multiple comparation was used for analysis. * *p* < 0.05; ** *p* < 0.01; *** *p* < 0.001.

**Figure 7 ijms-25-12315-f007:**
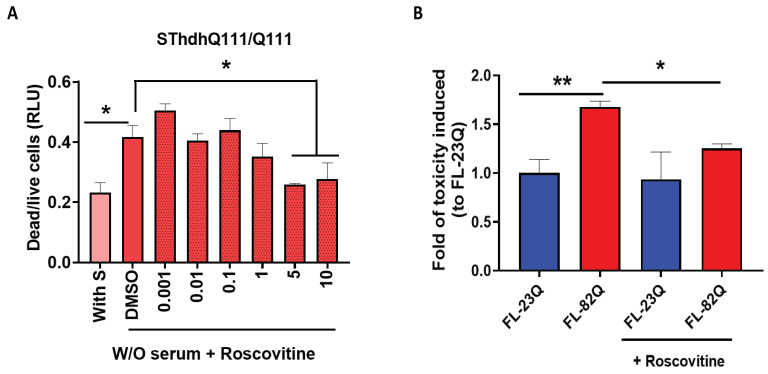
Roscovitine protected cells from mHTT-induced toxicity. Roscovitine was used in different HD cell models to evaluate its effect on mHTT-induced toxicity. (**A**). The effect of roscovitine on mHTT-induced toxicity in SThdh cells expressing Q111 cells. SThdh ^Q111/Q111^ cells were treated with roscovitine for 24 h under serum withdrawal conditions, and cell toxicity was measured using a CytoTox kit. (**B**). The effect of roscovitine on mHTT-induced toxicity in primary cortical neuronal HD cell model. Primary cultured cortical neurons were transiently transfected with plasmids expressing either normal Q (23Q) or poly Q (82Q) for 4 h and then treated with or without roscovitine for 48 h. A nuclei condensation assay was used to evaluate cell death. The experiment was repeated by at least two different analysts. Each experiment had *n* = 3. One representative experiment is presented. W/O = serum withdrawal. One-way ANOVA with Dunnett’s multiple comparation was used for analysis. * *p* < 0.05; ** *p* < 0.01.

**Figure 8 ijms-25-12315-f008:**
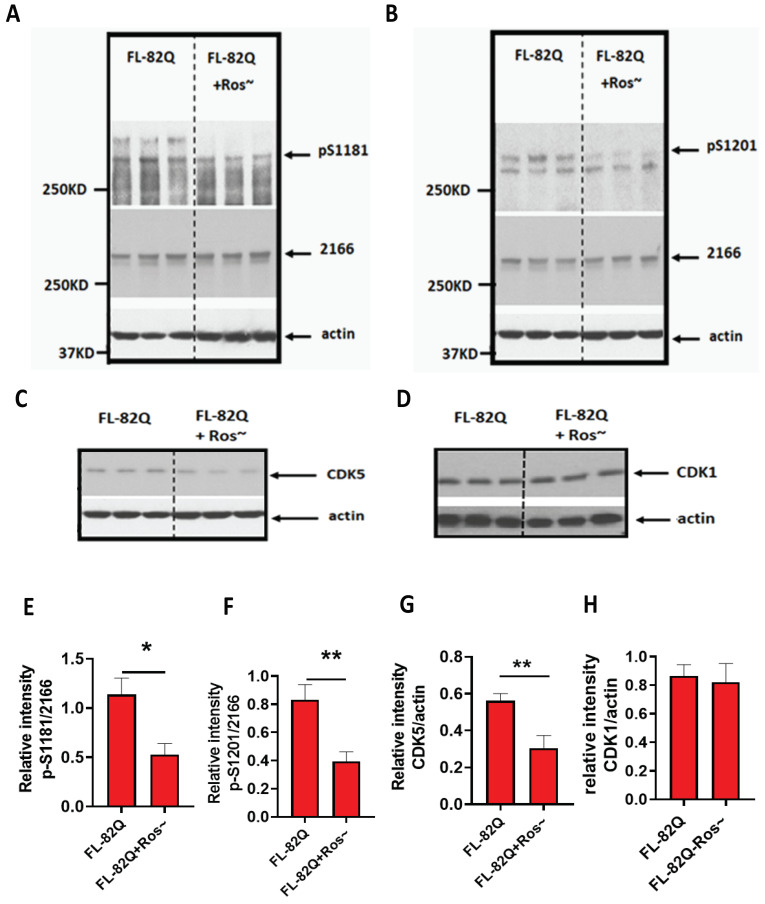
Roscovitine reduced the phosphorylation of S1181 and S1201 of mHTT. Roscovitine reduced p-S1181-HTT and p-S1201-HTT in vitro by inhibiting CDK5 but not CDK1. (**A**) The representative western blot for p-S1181 in HEK293 cells transfected with a plasmid expressing FL-82Q and treated with or without roscovitine for 48 h. (**B**) The representative western blot for p-S1201 in HEK293 cells transfected with a plasmid expressing FL-82Q and treated with or without roscovitine for 48 h. (**C**) The representative western blot for CDK5 in HEK293 cells transfected with a plasmid expressing FL-82Q and treated with or without roscovitine for 48 h. (**D**) The representative western blot for CDK1 in HEK293 cells transfected with a plasmid expressing FL-82Q and treated with or without roscovitine for 48 h. (**E**) Quantification of p-S1181 in A. (**F**) Quantification of p-S1201 in B. (**G**) Quantification of CDK5 in C. (**H**) Quantification of CDK1 in D. The experiment was repeated by at least two different analysts. Each experiment had n = 3. One representative experiment is presented. Ros~ = Roscovitine. A two-tailed Student *t*-test was used. * *p* < 0.05; ** *p* < 0.01.

**Figure 9 ijms-25-12315-f009:**
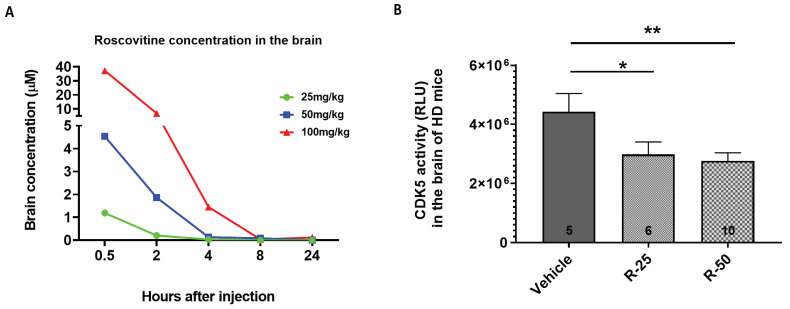
Roscovitine penetrated the mouse brain and inhibited CDK5 activity in the zQ175 HD mouse brain. Variant doses of roscovitine were injected into 4-month-old mice through intraperitoneal injection (IP). Plasma and brain samples were collected at different time points after injection. The concentration of roscovitine was measured by LCMS/MS. (**A**) Brain concentration of roscovitine at different time points after acute IP injection with variant doses. Each group has 4–5 mice. (**B**) The CDK5 activity in the brain of mice injected with roscovitine for three weeks. Two doses of roscovitine were injected into 4-month-old zQ175HD mice by IP daily for three weeks. Twenty-four hours after the last injection, the mouse brain samples were collected, and the CDK5 activity was measured using a commercial kit from Promega. Animal number is indicated in the bar graph for each group. R-25 = 25 mg/kg of roscovitine. R-50 = 50 mg/kg of roscovitine. One-way ANOVA with Fish’s LSD was used for analysis. * *p* < 0.05; ** *p* < 0.01.

**Figure 10 ijms-25-12315-f010:**
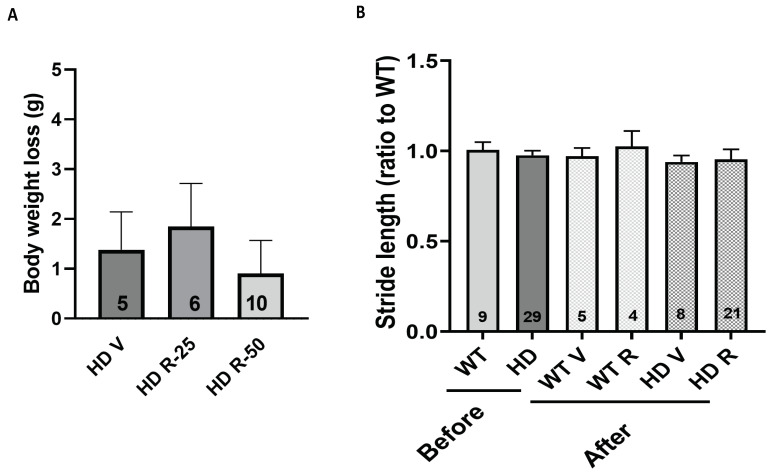
Tolerability of mice injected with roscovitine by IP administration. (**A**) The body weight loss of zQ175 HD mice injected with roscovitine. (**B**) The stride length (by CatWalk) in zQ175 HD mice before and after being injected with roscovitine. Roscovitine was injected by IP daily for three weeks. Animal number for each group is indicated in the bar graphs. V = Vehicle, R = roscovitine. R-25 = 25 mg/kg of roscovitine. R-50 = 50 mg/kg of roscovitine. One-way ANOVA with Fish’s LSD was used for analysis.

**Figure 11 ijms-25-12315-f011:**
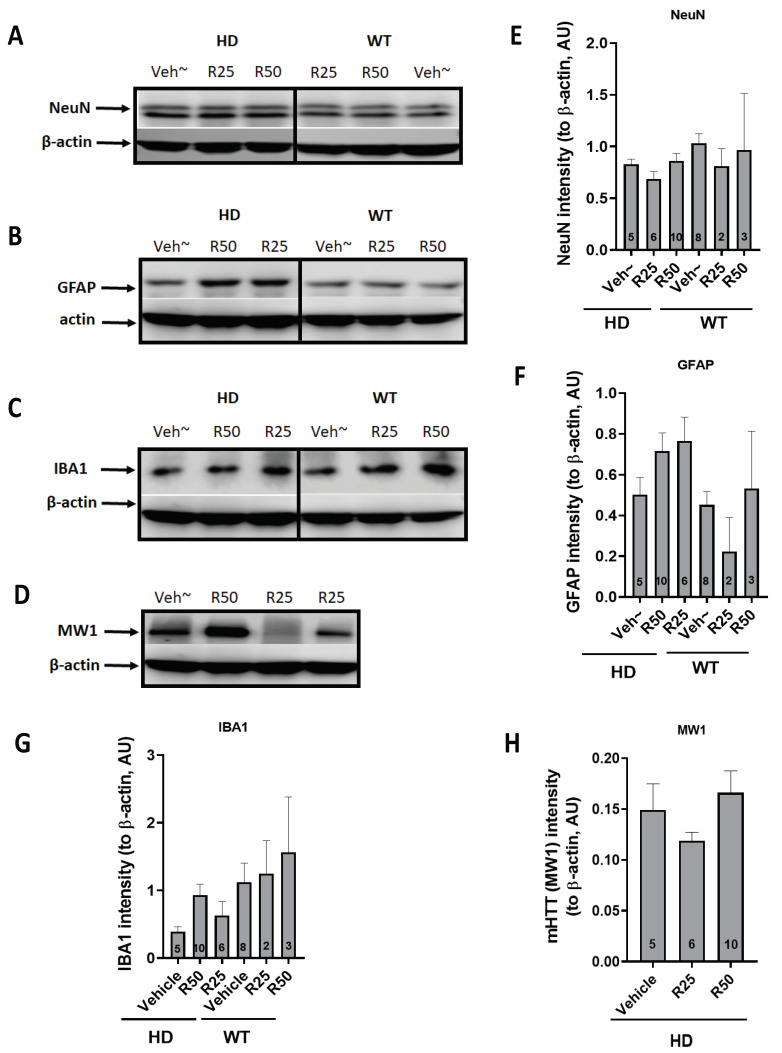
Molecular neurobiological measurement in the brain of zQ175 HD mice injected with roscovitine by IP. Brain samples were collected as previously described. The cortex was dissected, and lysates were made for western blot analysis. (**A**) The expression of a pan-neuronal marker, NeuN. (**B**) The expression of astrocyte marker GFAP. (**C**) The expression of microglial marker IBA1. (**D**) The expression of mHTT marker MW1. (**E**) The quantification of NeuN. (**F**) The quantification of GFAP. (**G**) The quantification of IBA1. (**H**) The quantification of MW1. Animal number is indicated in the bar graph for each group. Veh~ = Vehicle, R25 = 25 mg/kg of roscovitine. R50 = 50 mg/kg of roscovitine. One-way ANOVA with Fish’s LSD was used for analysis.

## Data Availability

All the data included in the present study can be shared upon request to the authors or upon request to the Data Repository of Johns Hopkins university.

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
