# Peer review of "Roscovitine, a CDK Inhibitor, Reduced Neuronal Toxicity of mHTT by Targeting HTT Phosphorylation at S1181 and S1201 In Vitro"

_ijms, 2024, doi:10.3390/ijms252212315_

Round 1
Reviewer 1 Report
Comments and Suggestions for Authors
In their paper, entitled “Titl Roscovitine, a CDKs inhibitor, reduced neuronal toxicity of mHTT by targeting HTT phosphorylation at S1181 and S1201 in vitro.”, the Authors describe the results of a study aimed at analyzing the role of phosphorylation in the toxicity of mutated huntingtin protein (mHTT), which causes the Huntington’s disease (HD), and at finding kinase inhibitors which can modulate toxicity. They report that, by reducing phosphorylation at the level of S1181 and S1201, neuronal toxicity induced by mHTT is also reduced. Moreover, the Authors report that Roscovitine, a CDKs inhibitor, able to reduce the p-S1181 and p-S1201, has a protective effect against mHTT toxicity. Most important, they found that roscovitine can cross the blood-brain barrier and penetrate the mouse brain, after intraperitoneal injection. Thus it can inhibit CDK5 activity in the brains of HD mice, thus suggesting that this inhibitor might be used for therapy also in humans.
The paper is of interest, also for the great variety of experimental approaches used. Thus, it is suitable for International Journal of Molecular Sciences, with only two very minor modifications:
1. Given the finding concerning cyclin-dependent kinases, and especially CDK5, a few words on cyclin production in the post-mitotic neurons might be of interest in both the Introduction and Discussion;
2. Page, 2, line 15: it should be better to write “The HTT gene contain 67 exons and encodes a large protein with a molecular weight of about 350 KD”.
Author Response
Please see the attachment. Response in Red font.

Reviewer 2 Report
Comments and Suggestions for Authors
In this manuscript, the authors investigated the potential of roscovitine, a CDK inhibitor, in mitigating the neuronal toxicity associated with mutant huntingtin protein (mHTT) in HD. Through in vitro and in vivo assays, the study identified CDK5 as a key kinase modulating mHTT toxicity via phosphorylation at specific serine sites (S1181, S1201). The findings suggest that roscovitine effectively reduces mHTT-induced toxicity by inhibiting CDK5 activity and thereby decreasing phosphorylation at these sites. This effect was demonstrated in cultured cells and in HD mouse models, where roscovitine showed the capability to cross the blood-brain barrier and reduce CDK5 activity without significantly altering body weight or inducing glial responses. Additionally, the authors note that high doses of roscovitine may be toxic to mice, underscoring the need for careful dose optimization. However, several experimental details require further clarification, and the data presented from animal studies are limited, making it challenging to fully assess the authors' hypothesis. Key information, such as brain pathology sections, mHTT aggregation changes, behavioral assessments, and survival data, is lacking, hindering a comprehensive evaluation of roscovitine's efficacy in this model. If additional animal data cannot be provided, I recommend reconsidering the inclusion of these in vivo results to ensure the findings presented are robust and conclusive. Some specific comments are attached below.
1. In the Introduction section, the authors are encouraged to provide additional background on cyclin-dependent kinase 5 (CDK5), particularly its role in neuronal function and potential mechanisms related to Huntington’s disease (HD) pathology. Including a summary of CDK5’s known contributions to HD pathogenesis and toxicity mechanisms would provide readers with a clearer foundation for understanding its significance in the study and how targeting CDK5 may offer neuroprotective effects against HD-associated neuronal toxicity.
2. Upon comparing Figures 1 and 2, it appears there is an inconsistency in the role of CDK5 in the phosphorylation of specific serine residues. In Figure 1, the screening results for S1201 and S2653 do not indicate significant involvement of CDK5, and instead, CDK1 seems to have a higher participation. However, in Figure 2, the authors emphasize the importance of CDK5 in phosphorylating residues 1181, 1201, and 2653. Could the authors clarify this discrepancy and provide additional insight into why CDK5’s role is highlighted in Figure 2 despite the seemingly minimal involvement suggested by the initial screening results in Figure 1? This clarification would enhance the consistency and interpretation of the findings.
3. In Figure 3, the knockdown efficiency of CDK5 appears to be modest, with protein expression inhibition not exceeding 50%. Given this relatively low knockdown level, the observed substantial reduction in phosphorylation raises concerns about potential off-target effects. Could the authors elaborate on the specificity of CDK5 knockdown in this context and discuss whether alternative mechanisms or off-target effects might account for the pronounced phosphorylation inhibition observed? This additional clarification would help in understanding the reliability and specificity of the knockdown results presented.
4. Figure 5: The study initially focused on mHTT-Q87, but there is an abrupt shift to mHTT-Q111 in the later analysis. In addition, given that mHTT aggregation is crucial for its pronounced cytotoxic effects, it is important to clarify why this transition was made and whether this affects the observed outcomes. Specifically, could the authors address whether CDK5 regulation influences the aggregation of mHTT-Q111? Elucidating this would be valuable for understanding the role of CDK5 in modulating mHTT’s aggregation-dependent toxicity.
5. Figure 6: In Figure 6, it would be insightful to know if the S to A mutations at the phosphorylation sites impact the aggregation behavior of mHTT in HEK293 cells. The inclusion of immunofluorescence staining results to visualize mHTT aggregation would significantly enhance the interpretation of these findings.
6. Figure 7 and 8: It is known that roscovitine inhibits CDK5 through competitive binding at the ATP site, blocking ATP interaction. However, roscovitine also inhibits other CDKs, such as CDK1 and CDK2, making it challenging to exclude potential involvement from these kinases. I recommend that the authors include additional CDK5-specific inhibitors to strengthen the evidence and improve the reliability of the findings related to CDK5’s role in this study.
7. Figure 9 presents data from an HD animal model; however, the information provided is quite limited. To enhance the robustness and interpretability of the findings, I recommend including additional essential data such as brain pathology sections, changes in mHTT aggregation, as well as basic information on animal behavior and Kaplan-Meier survival curves. These additions would provide a more comprehensive understanding of the effects observed in this model.
8. It is well-established that CDK5 cannot activate itself and requires binding with specific activator proteins to function. The primary activators, p35 and p39, form active complexes with CDK5, with p35 playing a crucial role in neurodevelopment. However, when p35 is cleaved into p25, it leads to CDK5 hyperactivation, which is implicated in neurodegenerative diseases. In this study, it is essential for the authors to specify whether the analyzed CDK5 activity is associated with p25 or p35. Clear differentiation and labeling are necessary to accurately interpret the implications of CDK5 activity in the experimental context.
Author Response

(The authors gave the same response as above.)

Reviewer 3 Report
Comments and Suggestions for Authors
In the study “Roscovitine, a CDKs inhibitor, reduced neuronal toxicity of mHTT by targeting HTT phosphorylation at S1181 and S1201 in vitro” by Liu et al., the authors test the effects of, a well-known CDK5 inhibitor – roscovitine on mHTT toxicity.
The study is interesting and paves the way towards potential novel approach in the treatment of Huntington’s disease. It merits publication but there are some issues that need clarification.
Introduction:
The results are explained in the introduction (the kinase screening test, etc). That should be omitted from the introduction section. Instead, more information about the mechanism of CDK action and the specific role of CDK5 and 1 in Huntington’s should be introduced.
It is unclear in which way is the role of CDK5 in the brain unique?
The authors state “In adults, it (CDK5) has a critical role in various neuronal functions [26-28, 32-34]. It is involved in regulating neuronal survival, synaptic plasticity, learning and memory formation, pain signaling, drug addiction, and long-term behavioral changes[35-37].” To what neuronal functions references 26-28 and 32-34 refer to, considering that they are obviously different than the neuronal functions explained in the references 35-37?
It will be helpful to the readers to explain the treatment potential of roscovitine in the Introduction section.
It is not customary to explain the results in the Introduction section. The authors can explain what they have done in the study without stating the outcomes.
Results:
General remarks – all the graphs should have individual values presented. All the legends should have number of cell culture experiments performed and number of animals used per experimental group.
It is mentioned in the Statistical analyses that all the analyses were done in triplicates. On what basis was the number of replicates chosen (G-power analyses)?
2.1.
From the results of the kinase assay presented it is not clear why are CDK5 and 1 picked for the study.
Presenting diagram depicting the location of specific serine phosphorylation sites in mHTT will be beneficial for the readers. The sequence of all the peptides used for the kinase assay should be presented in the Supplemental data file.
In the supplemental Figure 1B it should state S120/buffer and not S116S120/buffer.
Why is there three different labelings on the y-axes in Figure 1. ?
The authors state: “Both CDK1 and CDK5 had relatively higher activity in phosphorylation of the S1181 and S1201 peptides among all tested CDKs. “ Based on what is the quantification established?
2.4.
SThdhQ111/ Q111 cells, as well as SThdhQ7/ Q7 cells should be briefly explained in the text of the Result section 2.4.
2.5.
The primary neuronal HD cell model is first mentioned in the chapter 2.5. but it is explained in the chapter 2.6. In addition, there is no references for this model in the chapter 2.5. Why were two methods for quantifying cell toxicity used?
2.9. HD mice should be named zQ175 HD as it is mentioned once in the Discussion part of the MS. The authors state that there were no significant differences between vehicle and roscovitine treated WT and HD mice in terms of the expression of NeuN, GFAP, and IBA1. Yet, they attempt to further interpret slight trends in expression levels changes. Either the number of animals should be increased (did G-power analysis endorsed the results) in order to validate statistical significance or the lack of it, or the current interpretation of the results should be omitted.
The statements “We also noticed a decreased trend of MW1 expression in the low dose of roscovitine-injected HD mice (Figure 11D and H). However, it was surprising that 3 weeks of treatment could reduce the level of mHTT for diseases like HD in vivo. “ should be omitted as they are unclear.
Figure 11. the title of the legend is misleading – There is no neuropathology in either WT or HD brain according to the presented results.
Material and Methods
How many animals in total were used for the analyses of roscovitine concentration in the brain and plasma?
All the uM should be corrected.
Discussion
The first paragraph in the Discussion section would fit better in the Introduction section.
Reference 73 refers to the inhibition of CDK5 in ALS not in HD.
The repetition from the Introduction and Results part of the MS should be omitted in the Discussion.
Comments on the Quality of English Language
The use of English language needs some reviewing.
Author Response

(The authors gave the same response as above.)

Round 2
Reviewer 2 Report
Comments and Suggestions for Authors
All of my questions have been addressed by the authors. I don't have any more queries.
Reviewer 3 Report
Comments and Suggestions for Authors
I think that the authors have adequately addressed the comments made by the reviewers in the revised version of the manuscript. Therefore, I have no further comments.